# Redefining Machine Simultaneous Interpretation: From Incremental Translation to Human-Like Strategies

## Abstract

Simultaneous Machine Translation (SiMT) requires high-quality translations under strict real-time constraints, which traditional encoder-decoder policies with only READ/WRITE actions cannot fully address. We extend the action space of SiMT with four adaptive actions: **SENTENCE_CUT**, **DROP**, **PARTIAL_SUMMARIZATION** and **PRONOMINALIZATION**, which enable real-time restructuring, omission, and simplification while preserving semantic fidelity. We implement these actions in a decoder-only large language model (LLM) framework and construct training references through action-aware prompting. To evaluate both quality and latency, we further develop a latency-aware TTS pipeline that maps textual outputs to speech with realistic timing. Experiments on the ACL60/60 English–Chinese and English-German benchmarks show that our framework consistently improves semantic metrics (e.g., COMET-KIWI) and achieves lower delay (measured by Average Lagging) compared to reference translations and salami-based baselines. Notably, combining **DROP** and **SENTENCE_CUT** yields the best overall balance between fluency and latency. These results demonstrate that enriching the action space of LLM-based SiMT provides a promising direction for bridging the gap between human and machine interpretation.

## 1 Introduction

Simultaneous speech translation requires real-time translation with high quality, which poses unique challenges compared to offline machine translation (MT) due to the incompleteness of information. While traditional MT systems generate fluent and accurate translations by relying on complete source sentences, such a paradigm is unsuitable for simultaneous machine translation (SiMT), where incremental processing and low latency are mandatory. The central bottleneck of SiMT therefore lies in maintaining an optimal balance between translation quality and latency. To address this challenge, the system must be capable of deciding both *when* and *how* to translate under partial input.

A prospective approach is to learn from professional human interpreters, who strategically decide when to pause for more context while still conveying the essential meaning through rephrasing, summarization, or omission. For instance, the widely adopted *salami technique* breaks the sentences into minimal segments that contains enough information for translation. This segmentation effectively reduces long-distance word reordering caused by syntactic divergence across languages, while preserving semantic fidelity. By applying this technique to SiMT, the system benefits from the monotonicity and improved word-order alignment (Makinae et al., 2024).

Most existing SiMT systems are built upon encoder-decoder architectures, where the quality-latency trade-off is controlled by policies defined over a limited set of actions (READ/WRITE). Tremendous effort has been made to optimize the timing and choice of these actions, yet this limited action space are not capable to fully capture strategies that human interpreters apply, such as salami technique, partial summarization, appropriate omission and partial reordering. In contrast, decoder-only large language models (LLMs) are naturally capable of producing such proper modifications. However, their tendency to rely on full sentence often leads to offline-style translations, which violate the real-time constraint. This limitation can be alleviated by explicitly constraining the model with prompts tailored to simultaneous settings.

Another obstacle lies in training data. Most SiMT systems adopt offline translations as references. Although such translations are fluent and semantically faithful, they are unsuitable as references for SiMT because of different generation patterns. Training on them biases models toward waiting for complete input, thus increasing latency and contradicting the real-time requirement. As a result, it is crucial to generate high-quality reference interpretations that align with simultaneous interpretation patterns while preserving semantic fidelity.

In this study, we propose a decoder-only SiMT framework that introduces four novel adaptive actions in addition to READ and WRITE, namely **SENTENCE_CUT**, **PARTIAL_SUMMARIZATION**, **DROP** and **PRONOMINALIZATION**. These actions mimic human interpreter strategies and dynamically balance quality and latency. By introducing this expanded action space, we redefine machine simultaneous interpretation by shifting the focus from mere incremental translation to human-like strategies, while remaining firmly within the conventional SiMT setting. Our framework is based on decoder-only LLMs such as GPT-4o and Qwen3-8B, and we compare against strong baselines including salami-based segmentation, the recent LLM-based system TransLLaMA, and prompting strategies such as few-shot and dynamic in-context prompting. In addition, we develop latency-aware text-to-speech (TTS) pipeline based on word alignment and source timestamps, which enables realistic simulation of interpreter behavior and provides accurate measurements of latency (e.g., Average Lagging).

Our main contributions are as follows.[1]

- We propose a decoder-only SiMT framework introducing four novel actions—**SENTENCE_CUT**, **PARTIAL_SUMMARIZATION**, **DROP** and **PRONOMINALIZATION**—integrated into a sequential decision-making process to balance translation quality and latency.

- We adapt offline translations into SiMT–like references using these actions, and produce training data that better reflects real-time interpretation constraints while preserving semantic fidelity.

- We conduct a comparative analysis of various training and inference methods including TransLLaMA system, few-shot prompting and dynamic in-context learning on Qwen3-8B to identify effective approaches for SiMT.

- We developed a latency-aware TTS pipeline based on word alignment and source timestamps, enabling realistic simulation of interpreter behavior and synchronous evaluation of quality and latency.

## 2 RELATED WORK

Current SiMT systems are typically evaluated on both translation quality and latency. For quality, n-gram-based metrics such as BLEU (Papineni et al., 2001), chrF (Popović, 2015) and TER (Snover et al., 2006) remain widely used due to their simplicity and historical prevalence. They mainly capture surface-level overlap and may undervalue translations that differ in form but are still semantically valid. Neural-based evaluation metrics such as COMET (Rei et al., 2020; 2022) leverage pretrained multilingual encoders to compare meaning rather than form, and do not require reference translations at inference time. They have been shown to align better with human quality assessments (Glushkova et al., 2023), thus providing a more reliable measure for SiMT performance. For latency, commonly used measures include Average Lagging (AL) (Ma et al., 2019), Average Proportion (AP) (Cho & Esipova, 2016), and Differentiable Average Lagging (DAL) (Cherry et al., 2018). More recently, Average Token Delay (ATD) (Kano et al., 2023) has been proposed, which explicitly accounts for the end timings of partial translations and thus better reflects delays caused by long output segments. Although BLEU is widely used for translation quality evaluation, its reliance on offline references makes it unsuitable for SiMT, as it may increase latency when such patterns are followed. Thus, modifications of offline reference are proposed (Chen et al., 2020; Makinae et al., 2024; Zhao et al., 2021) to deal with the long-distance word reordering introduced by the word order differences between various languages.

In terms of architectural choices, most SiMT systems have been built on encoder–decoder frameworks, where the trade-off between quality and latency is controlled by a policy determining when

---

[1]Due to double-blind review, we do not release artifacts during the review phase. Upon acceptance, we will release the code and detailed prompts.

to READ (consume source tokens) and WRITE (generate target tokens). Policies can be broadly categorized into fixed, adaptive, and hybrid. Fixed policies include constant delay policies like wait-k (Ma et al., 2019) and read-m-write-n (Issam et al., 2024), and segmentation-based fixed policies like punctuation-based segmentation (Oda et al., 2014). Adaptive policies (Arivazhagan et al., 2019; Gu et al., 2016; Raffel et al., 2017; Oda et al., 2015; Zhao et al., 2021) dynamically adjust decisions based on the source content, model confidence, or predicted future context. The work of Oda et al. (2015) is particularly pioneering, as it selects actions by predicting unseen syntactic constituents using parser information, thus demonstrating one of the earliest syntax-informed approaches to simultaneous translation. Hybrid approaches combine the strengths of both: for instance, Adapters Wait-k & Adaptive Adapters (Issam et al., 2024) enable a single model to handle multiple wait-k settings while incorporating adaptive decision-making. Recently, efforts have also explored decoder-only models, such as the Decoder-only Streaming Transformer (Guo et al., 2024), Hibiki (Labiausse et al., 2025), and TransLLaMA (Koshkin et al., 2024), which reduce inference cost by discarding the encoder and integrate more naturally with large pretrained LLMs. In particular, TransLLaMA introduces the use of special <WAIT> tokens to synchronize source and target streams, ensuring that target words are only generated after sufficient source context becomes available.

Beyond architectures, model adaptation strategies play a key role. Early approaches adopted full fine-tuning (Luong & Manning, 12 3-4 2015; Freitag & Al-Onaizan, 2016; Sennrich et al., 2016), updating all parameters of pretrained models. However, this requires substantial amounts of task-specific data, which is scarce for SiMT (Zhang & Feng, 2023). To alleviate this, parameter-efficient fine-tuning (PEFT) methods have been introduced, enabling adaptation with limited data and resources. Among them, LoRA (Hu et al., 2021) is the most widely used and has been successfully applied in recent SiMT systems (Koshkin et al., 2024). At the same time, the rise of LLMs has brought interest in prompt-based adaptation. Few-shot prompting (Patel et al., 2023; Puduppully et al., 2023; Tang et al., 2025) demonstrates the ability of LLMs to mimic interpretation styles with only a handful of examples. Going further, dynamic in-context learning (DICL) (Rubin et al., 2022; Zhou et al., 2023) retrieves task-relevant examples at inference time and has shown strong results in general NLP, though it has not yet been applied to SiMT.

Distinct from previous work, we propose a SiMT framework based on decoder-only models empowered by four novel adaptive actions —**SENTENCE_CUT**, **PARTIAL_SUMMARIZATION**, **DROP** and **PRONOMINALIZATION** —to balance quality and latency by emulating the adaptive strategies of professional interpreters, which we describe below.

## 3    METHOD

Our proposed framework for decoder-only SiMT consists of four key components: (1) an extended action space that emulates human interpreter strategies, (2) model adaptation through LLM-based systems and prompting-based methods, (3) a latency-aware TTS pipeline, and (4) an inference procedure that performs step-by-step generation. The system overview is shown in Figure 1.

### 3.1    EXTENDED ACTION SPACE

Conventional SiMT policies are limited to two actions: READ and WRITE . Although the optimization of policies can improve quality-latency balance by making better decisions of when to commit WRITE actions, they cannot fully capture the techniques developed by human interpreters, which make translations more fluent and accurate. The salami technique (Makinae et al., 2024) adapts offline translations from MuST-C (Di Gangi et al., 2019) by splitting sentences into semantically sufficient segments, which improves word-order monotonicity and reduces latency.

Building on this idea, we generalize such human interpreter techniques into four new actions that can be dynamically invoked by LLMs during generation, enabling real-time application beyond static reference adaptation.

• **SENTENCE_CUT** Split long or syntactically complex clauses (e.g., relative clauses, appositives, inserted explanations) into shorter, grammatical sentence segments. For example, the sentence in ACL60/60 dataset *"At the position of at the pool party with Barack Obama, we got a graph with the right nodes on the person and the event subject, but guess the wrong timing information"* can be split after "with" since it connects two semantically complete parts. By inserting appropriate punctuation

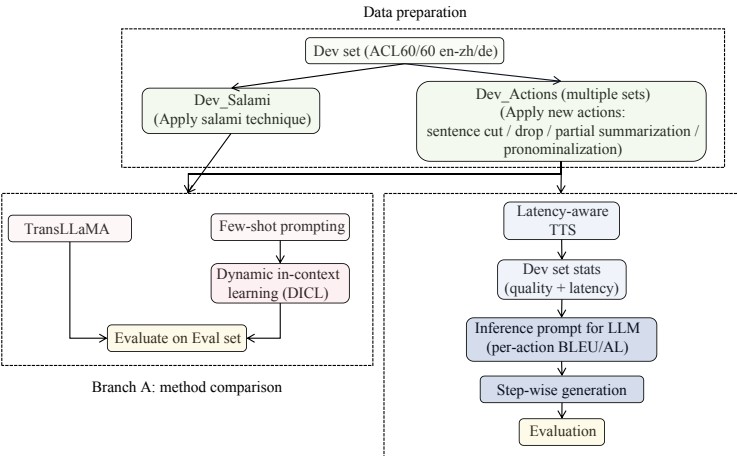

Figure 1: Method overview. We first construct development sets with either salami-based segmentation or our proposed action-based translations. Branch A compares different adaptation methods (LoRA fine-tuning, few-shot prompting, and dynamic in-context learning). Branch B integrates a latency-aware TTS pipeline to obtain quality and latency statistics, which are then used to guide inference through per-action prompts and step-wise generation.

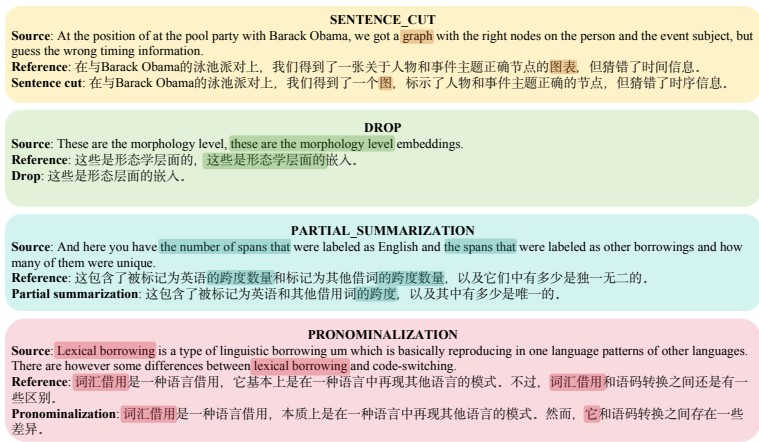

Figure 2: New actions and examples in en-zh SiMT. Each action is illustrated with an English source sentence, a literal reference translation, and an adapted version with the corresponding action applied. Highlighted spans indicate the parts of the sentence that are treated differently in the adapted translation compared to the reference (e.g., splitting, omission, summarization). This visualization shows how the original English segment is restructured, modified, or condensed in Chinese translation.

and connective words (which LLMs are able to supply), the sentence is divided into two fluent units, reducing reordering and improving latency.

- **DROP** Remove only truly non-informative content (e.g., "uh", "you know"), repeated words, or self-corrections. For instance, in *"These are the morphology level, these are the morphology level embeddings,"* the phrase *"these are the morphology level"* is repeated without adding new meaning. Applying **DROP** removes the redundancy and yields a cleaner, semantically accurate translation.

- **PARTIAL_SUMMARIZATION** Combine or simplify semantically equivalent or repetitive expressions while preserving the original meaning and tone (e.g., speculation, politeness). This is useful when multiple clauses convey essentially the same information. For example, in *"And here you have the number of spans that were labeled as English and the spans that were labeled as other borrowings and how many of them were unique,"* both clauses share the subject *"the number of spans."*

Summarization condenses the sentence into a more concise form, improving readability and reducing latency without loss of meaning.

- **PRONOMINALIZATION** Replace repeated or already mentioned noun phrases with pronouns only if referents are unambiguous. In two consecutive sentences "Lexical borrowing is a type of linguistic borrowing um which is basically reproducing in one language patterns of other languages. There are however some differences between lexical borrowing and code-switching", the phrase "lexical borrowing" is repeated. Since no ambiguity would be caused if we replace the second phrase into a pronoun like "it", and the two phrases are close enough to each other, we can apply **PRONOMINALIZATION** to convert this long phrase into a pronoun for more fluent expression.

Examples are shown in Figure 2 for illustration. These actions are applied at each decision point, and enables the model to adjust its behavior according to available context and the latency constraints. Training references are prepared by prompting GPT-4o to generate translations under different action combinations. Detailed experiement settings can be found in appendices A.2.

### 3.2 Model adaptation

To evaluate how different approaches adapt decoder-only LLMs to the SiMT scenario, we conduct a comparative study of four methods:

- **TransLLaMA** (Koshkin et al., 2024) is a policy-free framework in which a pre-trained decoder-only LLM is supervised on causally aligned source–target pairs with inserted `<WAIT>` tokens. This design enables the model to directly learn when to emit translations and when to wait for more context, without relying on an external policy.

- **Few-shot prompting** LLMs show fantastic capability to learn specific patterns or styles of generation from a few examples in the prompts (Brown et al., 2020; Reynolds & McDonell, 2021). By selecting representative examples from each set of reference translations of the development-sets, we can guide the LLM to learn from their approaches of interpretation.

- **Dynamic in-context learning** Based on few-shot prompting, we want to test if DICL can choose appropriate examples in all the reference translations that best match the current translation task and guide the LLM's translation pattern, which is inspired by this method's excellent performance in classification tasks (Rubin et al., 2022; Zhou et al., 2023). We applied retrieval-based DICL of two methods: (1) keyword extraction + sentence classification and (2) embedding clustering to search for the most suitable examples according to current input sentences. Implementation details are provided in Appendices A.4.

We use Qwen3-8B as the base model when doing inferences because of its efficiency for deployment and strong cross-lingual capabilities (Yang et al., 2025). The inputs are word sequences which are transcriptions of ACL60/60 development set, and the output translations are also text in target languages. This follows the setting in TransLLaMA experiments. All four methods are applied under three different development set configurations: (a) reference translations, (b) salami-based segmentation, and (c) action-based references. This design allows us to compare their relative effectiveness across multiple adaptation scenarios, analyzing both translation quality and latency.

### 3.3 Latency-aware TTS pipeline

To get the latency information for evaluation and inference, we developed a latency-aware TTS pipeline. Specifically, we follow the pipeline shown in Figure 3. It first applies Whisper large-v2 (Radford et al., 2022) to extract source word timestamps, then aligns source–target words with *SimAlign* (Sabet et al., 2020). Based on this alignment, `<WAIT>` tokens are inserted to enforce causal order and derive segment timetables, which specify when each target chunk should be spoken. Finally, target speech is synthesized with Cozyvoice 2 (Du et al., 2024) for Chinese or Tacotron 2 (Shen et al., 2017) for German. Full step-by-step details are provided in Appendix A.3.

### 3.4 Inference procedure

We use a text-only SiMT setting at inference: the input is the source word sequence, and the output is target-side text. From development-set translations and TTS runs under different action combina-

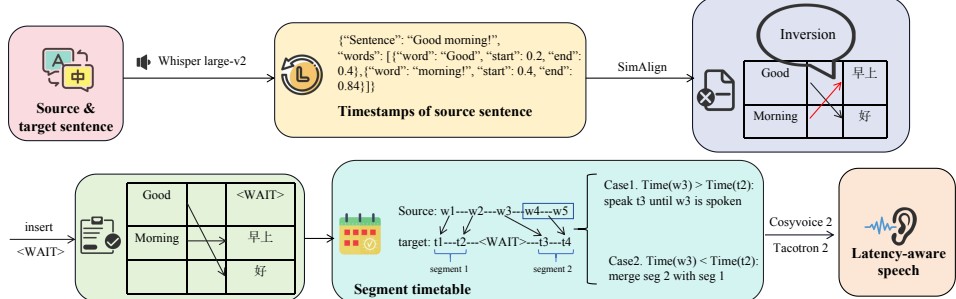

Figure 3: Latency-aware TTS. The system first obtains source word timestamps with Whisper and aligns source–target words using SimAlign. Special <WAIT> tokens are inserted to enforce causal alignment, which divides the target into segments. Each segment is then scheduled according to the corresponding source word timing and synthesized with CosyVoice2 (Chinese) or Tacotron 2 (German). This process produces speech outputs that reflect realistic latency for evaluation.

tions, we compute BLEU and AL and include these statistics in the inference prompt. Guided by these statistics, Qwen3-8B selects actions at each step to balance quality and delay. The prompt also specifies the simultaneous interpretation constraints and output format. The detailed prompt design can be found in appendices A.1.

## 4 EXPERIMENT AND ANALYSIS

### 4.1 DATA

For model adaptation, development-set (dev) action-combination sweeps, and inference, we use the ACL60/60 English-to-Chinese (en–zh) dataset (Salesky et al., 2023); we run the same protocol on the English-to-German (en–de) language pair. Throughout, the SiMT system operates in a text-only setting: the input is the English word-by-word transcript (i.e., a word sequence of the source sentence), and the output is the target-side text. The dev sets are augmented with GPT-4o reference translations covering different action combinations as well as the salami technique. These references are used both as LoRA training targets and as demonstration examples for prompting, and are further passed through our latency-aware TTS pipeline (aligned to source word timings) to obtain quality and latency statistics used at inference. Unless otherwise specified, final results are reported on the ACL60/60 evaluation (eval) sets.

### 4.2 EXPERIMENTAL SETUP

We used Qwen3-8B as the base model. For TransLLaMA-style supervised fine-tuning (SFT) (Koshkin et al., 2024), the training data were causally aligned by inserting <WAIT> tokens into the target side. We applied QLoRA with rank 16, $\alpha = 32$, and dropout 0.1, training for 2 epochs with AdamW (learning rate $5 \times 10^{-5}$, effective batch size 16). Training was performed on a single NVIDIA A100 80GB GPU.

### 4.3 EVALUATION METRICS

Translation quality was measured by BLEU, chrF, TER, and neural-based metrics COMET-da and COMET-KIWI. Latency was evaluated using Average Lagging (AL) only because it just serves as a reference of latency information in the inference stage.

### 4.4 MAIN RESULTS

**Model adaptation** We compare three adaptation strategies—TransLLaMA, few-shot prompting, and dynamic in-context learning (DICL)—under the salami, action-based, and reference translation settings. The evaluation results for ACL60/60 dev set are detailed in Table 1. Overall, DICL, especially the keyword-retrieval method, yields the best quality. It achieves the top or near-top scores

Table 1: Model adaptation under three supervision settings. We compare **TransLLaMA**, **Few-shot (static)**, and **Dynamic In-Context Learning** across: (a) *Salami-based* references , (b) *Action-adapted* references generated with the full action set, and (c) *ACL60/60* dev-set reference translations. We report surface-overlap metrics (BLEU/chrF/TER), semantic metrics (COMET-da/COMET-KIWI), and latency (AL, seconds↓).

(a) Salami-based references

| Method | BLEU | chrF | TER↓† | COMET-da | COMET-KIWI | AL (s)↓ |
|---|---|---|---|---|---|---|
| TransLLaMA | 57.66 | 41.36 | **96.72** | 0.8798 | 0.7950 | **0.813** |
| Few-shot (static) | 55.49 | 50.11 | 106.11 | 0.8779 | 0.7984 | 0.901 |
| DICL (keywords) | **60.31** | **54.50** | 103.71 | **0.8854** | **0.8046** | 0.891 |
| DICL (embedding) | 59.87 | 54.24 | 98.03 | 0.8847 | 0.8030 | 0.915 |

(b) Action-adapted reference generated with full action set

| Method | BLEU | chrF | TER↓ | COMET-da | COMET-KIWI | AL (s)↓ |
|---|---|---|---|---|---|---|
| TransLLaMA | **58.50** | 41.61 | **97.16** | 0.8816 | 0.8053 | **0.702** |
| Few-shot (static) | 55.80 | 50.31 | 105.90 | 0.8843 | **0.8080** | 0.857 |
| DICL (keywords) | 58.31 | **52.97** | 102.40 | **0.8869** | 0.8060 | 0.861 |
| DICL (embedding) | 57.44 | 52.26 | 98.03 | 0.8854 | 0.8037 | 0.867 |

(c) ACL60/60 dev-set reference

| Method | BLEU | chrF | TER↓ | COMET-da | COMET-KIWI | AL (s)↓ |
|---|---|---|---|---|---|---|
| TransLLaMA | **57.66** | **41.27** | **96.72** | 0.8852 | 0.8000 | **0.911** |
| Few-shot (static) | 55.79 | 39.11 | 110.77 | 0.8856 | **0.8079** | 0.916 |
| DICL (keywords) | 55.32 | 37.60 | 104.15 | 0.8822 | 0.8046 | 0.928 |
| DICL (embedding) | 57.17 | 40.97 | 101.53 | **0.8863** | 0.8058 | 0.919 |

† TER↓ is reported as a percentage, $100 \times \frac{\text{edits}}{|\text{ref}|}$, so values can exceed 100 when edits > reference length.

on BLEU/chrF and semantic metrics (COMET-da/COMET-KIWI) in salami and action setups, benefiting from retrieving input-relevant demonstrations. The embedding-based DICL is consistently close but slightly weaker. Few-shot (static) occasionally leads on COMET-KIWI but is less stable and more sensitive to example selection. TransLLaMA remains a strong latency-oriented method that delivers the lowest AL (notably 0.702s in Action) and often the best TER, making it preferable when minimal delay is desired. We also observe a metric split by supervision style: Salami-based data favors surface-overlap metrics (BLEU/chrF/TER), whereas action-adapted references improve semantic metrics (COMET). This aligns with our motivation that an enriched action space better preserves meaning under simultaneity. In summary, DICL with keyword retrieval offers the best overall quality and robustness under limited data, while TransLLaMA is the choice for strict latency constraints.

**Different choices of actions** To analyze the effect of individual and combined actions, we generated seven sets of translations on the ACL60/60 development set by prompting GPT-4o to perform step-by-step translation with different action combinations. The results in Table 2 show that individual actions like **SENTENCE_CUT**, **DROP**, and **PARTIAL_SUMMARIZATION** each brings moderate improvements in fluency or latency, but combining them leads to more significant gains. For en-zh, the combination **DROP + SENTENCE_CUT** achieved the lowest latency (0.817s), while using all actions together yielded the highest BLEU and COMET scores. For en-de, similar patterns were observed: **SENTENCE_CUT + DROP** minimized latency (0.252s), whereas combining multiple actions improved semantic fidelity (highest COMET-KIWI). These results confirm that different actions complement each other in balancing quality and latency.

**Inference** We provided the quality (BLEU) and latency (AL) scores of each action obtained on the dev sets for Qwen3-8B and instructed it to choose an appropriate action at each step during interpretation. The comparisons of inference results and outputs obtained by applying the salami technique versus action-based inference on the eval sets of both English–Chinese (en-zh) and English–German (en-de) are reported in Table 3.

Table 2: Performance of different action combinations on the dev set. Note that we mark the best results of each metric using **bold** letters and the best results except for the reference translations using pink highlight .

| EN-ZH Translation Performance | | | | | | |
|---|---|---|---|---|---|---|
| **Action Combination** | **BLEU** | **chrF** | **TER↓** | **COMET-da** | **COMET-KIWI** | **AL (s)↓** |
| Salami only | 57.26 | 38.83 | 104.73 | 0.8567 | 0.7727 | 0.825 |
| SENTENCE_CUT | 60.28 | 53.99 | 101.58 | 0.8765 | 0.7927 | 0.824 |
| DROP | 58.94 | 52.69 | 101.18 | 0.8733 | 0.7909 | 0.851 |
| PARTIAL_SUMMARIZATION | 60.33 | 53.67 | 98.22 | 0.8764 | 0.7923 | 0.847 |
| PRONOMINALIZATION | 60.85 | 41.39 | 101.78 | 0.8738 | 0.7910 | 0.858 |
| SENTENCE_CUT + DROP | 60.67 | 41.88 | 102.37 | 0.8745 | 0.7911 | **0.817** |
| DROP + PARTIAL_SUMMARIZATION + PRONOMINALIZATION | 59.91 | 53.43 | 98.22 | 0.8764 | 0.7924 | 0.888 |
| All actions | 62.67 | 46.28 | 99.80 | 0.8944 | 0.7952 | 0.922 |
| ACL60/60 ref | **100.00** | **100.00** | **0.00** | **0.9549** | **0.7983** | 0.972 |
| EN-DE Translation Performance | | | | | | |
| **Action Combination** | **BLEU** | **chrF** | **TER↓** | **COMET-da** | **COMET-KIWI** | **AL (s)↓** |
| Salami only | 47.48 | 69.86 | 38.94 | 0.8534 | 0.8102 | 0.284 |
| SENTENCE_CUT | 44.05 | 69.87 | 42.66 | 0.8525 | 0.8076 | 0.317 |
| DROP | 44.90 | 68.61 | 42.63 | 0.8442 | 0.7988 | 0.358 |
| PARTIAL_SUMMARIZATION | 45.05 | 69.31 | 41.73 | 0.8581 | 0.8086 | 0.361 |
| PRONOMINALIZATION | 44.96 | 69.40 | 41.89 | 0.8505 | 0.8074 | 0.352 |
| SENTENCE_CUT + DROP | 44.74 | 69.18 | 41.93 | 0.8501 | 0.8068 | **0.252** |
| DROP + PARTIAL_SUMMARIZATION + PRONOMINALIZATION | 44.95 | 69.19 | 42.08 | 0.8542 | **0.8198** | 0.261 |
| All actions | 44.88 | 69.11 | 42.05 | 0.8526 | 0.8082 | 0.253 |
| ACL60/60 ref | **100.00** | **100.00** | **0.00** | **0.9549** | 0.7983 | 0.921 |

Across both language pairs, inference guided by action choices consistently outperforms salami segmentation in most quality metrics, and even yields COMET-KIWI scores comparable to the reference translations. Notably, the action combination "**DROP + SENTENCE_CUT**", which achieved the lowest latency on the dev sets, also leads to the best trade-off on the eval sets. For en-zh, it delivers the highest BLEU and COMET-KIWI while substantially reducing AL. For en-de, it improves all quality scores simultaneously while maintaining competitive latency.

These results suggest a shared trend across both language pairs: action-aware inference is better at balancing semantic fidelity and latency than salami-based segmentation. The consistent advantage of "**DROP + SENTENCE_CUT**" can be explained by their complementary effects—**DROP** removes redundant or filler material, reducing delay, while **SENTENCE_CUT** alleviates long-distance re-ordering by segmenting complex clauses. Together, they enable translations that are both more fluent and faster, aligning with human interpreter strategies.

For example, given the source sentence "In other words, it cannot be used for many projects on GitHub", the translation obtained with salami segmentation was "换句话说, 它不能被使用在许多项目中在 GitHub 上" (BLEU 49.25), where "GitHub 上" was placed at the end, resulting in an expression that does not conform to natural Chinese usage and providing little benefit in latency reduction. In contrast, inference with **CUT+DROP** generated "换句话说, 它不能用于 GitHub 上的许多项目" (BLEU 100.00), which is both fluent and faithful. As another case, for the sentence "For those samples without unused quantities, so the overall performance is actually higher than the, the performance is actually higher than the overall performance", the salami-based output preserved the redundant phrase (BLEU 52.61), while the **DROP** action effectively removed it, yielding "对于那些没有未使用数量的样本, 所以整体性能实际上高于整体性能" (BLEU 66.89). This not only improved translation quality but also reduced latency since the output was shorter.

Table 3: Performance of different inference strategies on the eval set.

(a) English-Chinese (en-zh)

| Method | BLEU | chrF | TER↓ | COMET-da | COMET-KIWI | AL (s)↓ |
|---|---|---|---|---|---|---|
| Salami | 57.21 | 40.57 | 110.04 | 0.8705 | 0.7846 | 0.802 |
| Inference by choosing actions | 62.44 | 46.80 | 126.20 | 0.9002 | 0.8020 | 0.814 |
| Inference by choosing from DROP / SENTENCE_CUT | 62.84 | 44.06 | 104.80 | 0.8891 | 0.8040 | 0.772 |
| ACL60/60 ref | 100.00 | 100.00 | 0.00 | 0.9582 | 0.8029 | 0.972 |

(b) English-German (en-de)

| Method | BLEU | chrF | TER↓ | COMET-da | COMET-KIWI | AL (s)↓ |
|---|---|---|---|---|---|---|
| Salami | 47.48 | 69.87 | 38.94 | 0.8534 | 0.8102 | 0.385 |
| Inference by choosing actions | 47.80 | 70.08 | 38.72 | 0.8541 | 0.8108 | 0.293 |
| Inference by choosing from DROP / SENTENCE_CUT | 49.97 | 70.96 | 37.38 | 0.8594 | 0.8111 | 0.357 |
| ACL60/60 ref | 100.00 | 100.00 | 0.00 | 0.9511 | 0.8048 | 0.921 |

## 4.5 ADAPTIVE BEHAVIOR STUDY

To further test whether the LLM adapts its decisions according to provided statistics, we modified the BLEU and AL scores of one action in the inference prompt. By artificially increasing its BLEU and lowering its AL, the model was encouraged to prefer this action (in our case, **PARTIAL_SUMMARIZATION** and **SENTENCE_CUT**). As expected, the LLM adjusted its behavior and invoked these actions more frequently during inference. The resulting translations exhibited a clear reduction in AL, while quality metrics only decreased marginally. This finding highlights that the model does not simply follow static templates, but is able to make data-driven decisions to balance quality and latency. The experiment details can be found in A.7.

## 5 CONCLUSIONS

In this work, we presented a decoder-only SiMT framework that extends the conventional READ/WRITE paradigm with four adaptive actions: **SENTENCE_CUT**, **PARTIAL_SUMMARIZATION**, **DROP** and **PRONOMINALIZATION**. These actions simulate strategies used by professional human interpreters, and allow the LLMs to dynamically balance the quality-latency trade-off. We further developed a latency-aware TTS pipeline, which provides realistic delay measurements and enables synchronous evaluation of both quality and latency. Our experiments on the ACL60/60 benchmark show that the proposed approach yields consistent improvements over the salami-based segmentation. In particular, dynamic in-context learning (DICL) demonstrates strong performance, while action-specific strategies such as **DROP + SENTENCE_CUT** achieve the best overall trade-off between semantic fidelity and latency. Conceptually, our results chart a path for redefining machine simultaneous interpretation: from viewing SiMT as incremental token emission to treating it as real-time application of human-like strategies—while remaining firmly within the SiMT setting.

## 6 LIMITATIONS AND FUTURE WORK

While our TTS pipeline enables quantitative latency analysis, we have not yet leveraged its synthesized speech for human listening studies, which remain essential for perceptual evaluation. In addition, the current TTS timing policy differs from human interpreters: we trigger speech after each aligned *word* rather than after a complete *semantic unit*, which tends to yield lower AL than human practice. However, because all methods share the same protocol, any bias is constant across systems and does not affect conclusions about relative quality–latency trade-offs. Finally, to better align with end-to-end speech-to-speech SiMT, future work should move beyond word-sequence inputs and build frameworks that consume raw speech as input.

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

# A APPENDIX

## A.1 PROMPT OF ACTIONS

The following is an example of inference prompt letting the LLM choose from all the actions.

```
You are a simultaneous translation(en-zh) agent. Your task is to read
a source sentence word by word, and decide what action to take at each
step to optimize the balance between translation quality and latency.
Keep to the original meaning and word order of the sentence when doing
translation You can choose from the following actions: - READ: Wait for
the next source word (default). - WRITE: Output a target word or phrase.
- DROP: Remove previously read word(s) if they are meaningless fillers
(e.g., ``uh", ``um"), repetitions, false starts, or self-corrections. Use
only when clearly justified. - PARTIAL_SUMMARIZATION: Merge or simplify
redundant or equivalent expressions, while preserving the meaning and tone
(e.g., politeness, speculation). - CUT: Intentionally split the sentence
into two shorter, independently translatable units. Use only when the
sentence is long or syntactically complex. - PRONOUN: Replace a repeated
noun phrase with a pronoun ONLY IF the referent is unambiguous. Keep to
the original word order and meaning, and do the new actions only if it
considerably improve the latency or quality of interpretation. Based on dev
set evaluation: - DROP → AL ≈ 0.851s, BLEU ≈ 58.94 - PARTIAL_SUMMARIZATION
→ AL ≈ 0.847s, BLEU ≈ 60.33 - CUT → AL ≈ 0.824s, BLEU ≈ 60.28 - PRONOUN
→ AL ≈ 0.858s, BLEU ≈ 60.85 Only use DROP, PARTIAL_SUMMARIZATION, or
CUT if they reduce latency without hurting translation quality. --- You
will receive the full source sentence. Your job is: 1. Simulate the
step-by-step translation process internally; 2. Carefully choose the action
to take at each step **strictly based on the statistics provided above**; 3.
Output: action sequence of every step, explanation of choosing each action,
and the full translation of the sentence. 3. You are given only the prefix
of the source. DO NOT use any information beyond the current prefix. If you
find yourself relying on unseen future words, output the token <VIOLATION>
and stop. Source sentence:<input sentence>
```

To verify that our setting does not exploit unseen future tokens, we conducted a prefix-feeding sanity check. For a source sentence $x = (x_1, x2, ..., x_n)$, we iterate $t = 1$ to $n$. At step $t$, the model receives only the prefix $x_{1:t}$ under the same instruction template as our main prompt; it outputs one action from READ, WRITE, DROP, CUT, PARTIAL_SUMMARIZATION, PRONOMINALIZATION. If an outputting action is chosen (e.g., WRITE or PARTIAL_SUMMARIZATION), the model must emit an *incremental* target fragment. Crucially, previously emitted target tokens are immutable: later steps may append but never revise earlier output, i.e., the target stream is prefix-monotonic.

**Instruction.** We append the following constraint to the end of the main prompt:

```
You are a simultaneous translation(en-zh) agent. Your task is to read
a source sentence word by word, and decide what action to take at each
step to optimize the balance between translation quality and latency.
Keep to the original meaning and word order of the sentence when doing
translation You can choose from the following actions: - READ: Wait for
the next source word (default). - WRITE: Output a target word or phrase.
- DROP: Remove previously read word(s) if they are meaningless fillers
(e.g., ``uh", ``um"), repetitions, false starts, or self-corrections. Use
only when clearly justified. - PARTIAL_SUMMARIZATION: Merge or simplify
redundant or equivalent expressions, while preserving the meaning and tone
(e.g., politeness, speculation). - CUT: Intentionally split the sentence
into two shorter, independently translatable units. Use only when the
sentence is long or syntactically complex. - PRONOUN: Replace a repeated
noun phrase with a pronoun ONLY IF the referent is unambiguous. Keep to
the original word order and meaning, and do the new actions only if it
considerably improve the latency or quality of interpretation. Based on dev
set evaluation: - DROP → AL ≈ 0.851s, BLEU ≈ 58.94 - PARTIAL_SUMMARIZATION
→ AL ≈ 0.847s, BLEU ≈ 60.33 - CUT → AL ≈ 0.824s, BLEU ≈ 60.28 - PRONOUN
→ AL ≈ 0.858s, BLEU ≈ 60.85 Only use DROP, PARTIAL_SUMMARIZATION, or CUT
if they reduce latency without hurting translation quality. --- You will
receive **a word at one time** Your job is: 1. Simulate the step-by-step
translation process internally; 2. Carefully choose the action to take at
each step **strictly based on the statistics provided above**; 3. Output:
At each step, output the action you chose and the incremental translation.
If you choose READ or other actions that don't yield a translation, do
not output the translation. Just give me the action. When given the
complete sentence, output the whole sentence based on previous incremental
translations. You are not allowed to modify or overwrite your previous
output, only incremental translations are allowed.
```

**One-sentence demonstration.** Source: "The method works well for the cases where long inputs are considered." t=4 (prefix "The method works well"): action=WRITE → "该方法运行良好"; t=6 ( "The method works well for the cases" ): action=READ (no output); t=8 ( "···for the cases where long" ): action=CUT → append "，尤其适用于"; t=11 ( "···where long inputs are considered" ): action=WRITE → append "长输入的情形。" Final concatenation (end of sentence): "该方法运行良好，尤其适用于长输入的情形。"

**Finding.** Running this prefix-feeding procedure with the same template and decoding settings produces translations that are nearly identical to those obtained with the single-shot prompt used in our main experiments (differences are limited to minor punctuation or phrasing). We did not observe evidence of future-token leakage: the incremental fragments at step $t$ remain stable when we randomize the unseen suffix $x_{1:n}$, and the final full-sentence outputs match the single-shot results up to negligible surface variation.

## A.2 BATCH GENERATION FROM GPT-4O

We generate action-controlled SiMT outputs under a unified, reproducible pipeline. Inputs are line-delimited English sentences that are trimmed, deduplicated, and split into `.jsonl` shards; each sample is assigned a non-reversible hash as `custom_id` for idempotency and result alignment. Each `.jsonl` line specifies a `/v1/chat/completions` call with an identical prompt template that en-

forces online-style translation (final translation only, with only specified actions allowed, minimal long-distance reordering), fixed decoding and randomness controls (seed, temperature/top-$p$, max tokens), and `response_format=json_object` for structured parsing. Shards are submitted as independent batch jobs. Determinism is maintained by fixing seeds, model and dependency versions, the prompt template, and a stable write order after deduplication; outputs are merged and de-duplicated by `custom_id` before scoring. All methods share the exact same inputs, prompt, decoding parameters, and post-processing, ensuring that any systematic bias from the measurement pipeline is constant across systems and suitable for reliable *relative* latency and quality comparison.

### A.3 Latency-aware TTS pipeline

1. Apply Whisper large-v2 (Radford et al., 2022) to get timestamps of each English word in the source sentence.

2. Find best word level alignment between source and target sentences with *SimAlign* (Sabet et al., 2020).

3. Insert `<WAIT>` tokens before the target words if it appears before corresponding source words. In this way, we form causal alignment where the target words are never spoken before the source words.

4. Get segment timetables for target sentences. Specifically, `<WAIT>` tokens divide the sentences into segments, and the starting time to say each segment is decided by the starting time of the source word corresponding to the first word in this segment (represented by W). Two situations may happen: the source word is spoken *before* or *after* the previous word in target sentence was spoken. In the former case, the succeeding segment can be merged to the previous segment, while in the latter case, the succeeding segment should be spoken when W is spoken.

5. Synthesize speech using the segment timetables and merge them into a whole sentence with Cozyvoice2 (Du et al., 2024).

### A.4 Dynamic in-context learning

As stated in the main body, we applied retrieval-based DICL with two methods:

- **Keyword extraction + sentence classification**: We build a category-based few-shot library by classifying English–Chinese sentence pairs with keyword matching. This enables the model to retrieve examples that are more directly relevant to the current input.

- **Embedding clustering**: We embed English–Chinese sentence pairs with the `all-MiniLM-L6-v2` SentenceTransformer model (Reimers & Gurevych, 2020; Wang et al., 2020), then apply K-means clustering to group semantically similar pairs. At inference, the model retrieves examples from the cluster most similar to the current input.

### A.5 Latency evaluation

To evaluate latency at the speech level, we adapt the standard Average Lagging (AL) metric into a time-based version measured in seconds. The inputs are the source English speech segmented into words with end times $t_1, t_2, \ldots, t_{|X|}$ (from Whisper word-level alignment) and the target Chinese speech synthesized with TTS and re-aligned using Whisper, which provides timestamps $\tau_1, \tau_2, \ldots, \tau_{|Y|}$ for each generated unit (word or character).

We define $g(t)$ as the number of source words whose end times are earlier than or equal to the start time of the $t$-th target unit:

$$g(t) = |\{j : t_j \leq \tau_t|.$$

The ratio of target to source length is $\gamma = \frac{|Y|}{|X|}$. We further denote $\tau^* = \min t : g(t) = |X|$ as the first step at which all source words have been covered.

For each target step $t \leq \tau^*$, both the policy index $g(t)$ and the diagonal index $\frac{t-1}{\gamma}$ are projected back to the time axis using linear interpolation over $t_j$, denoted as time$(g(t))$ and time$((t-1)/\gamma)$. The

---

**Algorithm 1:** Time-based Average Lagging ($\text{AL}_{\text{sec}}$)

---

**Input:** Source word end times $\{t_1, \ldots, t_{|X|}\}$ (monotonic); target unit onset times $\{\tau_1, \ldots, \tau_{|Y|}\}$

**Output:** $\text{AL}_{\text{sec}}$

$\gamma \leftarrow |Y|/|X|$;

**for** $t \leftarrow 1$ **to** $|Y|$ **do**

    $g(t) \leftarrow \big| \{\, j \mid t_j \leq \tau_t \,\} \big|$;                 `// # source words finished by` $\tau_t$

$\tau^* \leftarrow \min\{\, t \mid g(t) = |X| \,\}$;

**if** *no such $t$* **then**

    $\tau^* \leftarrow |Y|$

**Define** TimeAtIndex$(x;\ t_1, \ldots, t_{|X|})$ as:

    **if** $x \leq 1$ **then return** $t_1$; **if** $x \geq |X|$ **then return** $t_{|X|}$;

    $i \leftarrow \lfloor x \rfloor$; $\ w \leftarrow x - i$;

    **return** $(1 - w)\, t_i + w\, t_{i+1}$;

$s \leftarrow 0$;

**for** $t \leftarrow 1$ **to** $\tau^*$ **do**

    $x_{\text{pol}} \leftarrow \max(1, \min(|X|,\ g(t)))$;

    $x_{\text{diag}} \leftarrow \max(1, \min(|X|,\ (t-1)/\gamma))$;

    policyTime $\leftarrow$ TimeAtIndex$(x_{\text{pol}};\ t_1, \ldots, t_{|X|})$;

    diagTime $\leftarrow$ TimeAtIndex$(x_{\text{diag}};\ t_1, \ldots, t_{|X|})$;

    $s \leftarrow s + (\text{policyTime} - \text{diagTime})$;

**return** $\text{AL}_{\text{sec}} \leftarrow s/\tau^*$;

---

time-based AL is then defined as

$$\text{AL}_{\text{sec}} = \frac{1}{\tau^*} \sum_{t=1}^{\tau^*} \left[ \text{time}\big(g(t)\big) - \text{time}\Big(\tfrac{t-1}{\gamma}\Big) \right].$$

This metric measures, in seconds, how much later the system commits target units compared with an ideal policy that follows the diagonal perfectly. In practice, we take the English word timestamps from Whisper as the reference timeline, the target emission times from TTS followed by Whisper alignment, and apply linear interpolation to map fractional token indices to real-valued source times. This ensures that AL reflects the true temporal delay rather than token-level alignment alone, making it a more faithful measure for speech-based simultaneous translation.

A.6    STEP-BY-STEP GENERATION OF ACTION SEQUENCE

We display examples of outputs of LLM that generate step-by-step action choices and explanation.

**Example 1** See action sequence in Table 4.

*Source*: We design our experiment as leave one out ah evaluation where we train FeSTe over sixteen datasets and apply it to the seventeenth dataset.

*Translations*: (Ours) 我们将实验设计为留一法评估, 其中我们在十六个数据集上训练 FeSTe, 并将其应用于第十七个数据集。[BLEU: 61.31]

(Salami) 我们设计了我们的实验, 作为留一法评估, 在这里我们训练 FeSTe 在十六个数据集上, 并将其应用于第十七个数据集。[BLEU: 38.58]

**Example 2** See action sequence in Table 5.

*Source*: Or in other words, we trained the language model to classify an abstract and class ah to abstract and class ah, if the abstract belongs to the class or not.

*Translations:* (Ours) 换句话说, 我们训练了语言模型, 将摘要和类别分类, 无论摘要属不属于类。[BLEU: 53.67]

| Step | Source word | Action | Output (if have) | Explanation |
|---|---|---|---|---|
| 1 | We | READ | — | Wait for the predicate; subject alone is incomplete. |
| 2 | design | READ | — | Likely a "design …as" frame; need object and complements. |
| 3 | our | READ | — | Part of the object NP. |
| 4 | experiment | WRITE | 我们将实验设计 | Core clause "we design our experiment" is complete; emit core without preempting "as". |
| 5 | as | WRITE | 为 | "design …as" is now confirmed; add "为". |
| 6 | leave | READ | — | Part of a fixed term. |
| 7 | one | READ | — | Part of a fixed term. |
| 8 | out | WRITE | 留一法 | "leave-one-out" fully recognized; safe to emit early. |
| 9 | ah | DROP | — | Clear filler; drop to reduce latency. |
| 10 | evaluation | WRITE | 评估 | Completes the term "leave-one-out evaluation". |
| 11 | where | WRITE | ，其中 | Clause linker; output for fluency. |
| 12 | we | READ | — | Wait for predicate and object. |
| 13 | train | READ | — | Wait for object and prepositional phrase. |
| 14 | FeSTe | READ | — | Needs the following PP before emitting. |
| 15 | over | READ | — | Preposition observed; needs its object. |
| 16 | sixteen | READ | — | Wait for the head noun. |
| 17 | datasets | WRITE | 我们在十六个数据集上训练 FeSTe | VP complete with PP; output in Chinese word order. |
| 18 | and | READ | — | Could coordinate a verb or NP; hold to avoid guessing. |
| 19 | apply | WRITE | ，并 | Seeing "apply" confirms coordinated verb; output "，并". |
| 20 | it | PRONOUN | 将其应用 | Pronoun refers to FeSTe; emit core predicate "应用" only; defer the "于". |
| 21 | to | WRITE | 于 | "to" arrives; add "于" to complete "应用于". |
| 22 | the | READ | — | Determiner; no emission needed. |
| 23 | seventeenth | READ | — | Wait for the noun to complete the phrase. |
| 24 | dataset | WRITE | 第十七个数据集。 | Target NP complete; finalize the sentence. |

Table 4: Simultaneous translation actions with Chinese outputs and English explanations. Textual output of LLM: Here's my simultaneous translation decision trace (word-by-word), choosing special actions only when they improve latency/quality per the given stats (PRONOUN best BLEU ≈60.85; PARTIAL_SUM ≈60.33 with low AL; DROP helps latency with minimal loss)

(Salami) 换句话说，我们训练了语言模型来分类一个摘要，并将类别 ah 分类为摘要和类别 ah，判断该摘要是否属于该类别。[BLEU: 32.16]

## A.7 Adaptive behavior

In order to examine whether the model adapts its action choices according to the statistics provided in the inference prompt, we designed an additional experiment focusing on sentences with the longest AL.

We first selected the 100 sentences from the eval set that yielded the longest AL under step-by-step inference with the original prompt. For each of these sentences, we then modified the inference prompt by artificially changing the statistics of one action:

In two separated experiments, we raised the BLEU score and lowered the AL of **PARTIAL_SUMMARIZATION** and **SENTENCE_CUT** respectively.

This manipulation effectively encouraged the model to prefer the corresponding action, while keeping all other settings unchanged.

Table 6 reports the results. In both cases, the model indeed invoked the promoted action more frequently. Consequently, the translations achieved a significant reduction in AL compared with the baseline, while semantic-based quality metrics decreased only marginally. These results clearly show that the LLM does not merely follow a fixed template, but actively adjusts its behavior in response

| Step | Source word | Action | Output (if have) | Explanation |
|------|-------------|--------|------------------|-------------|
| 1 | Or | READ | — | Connector starts a phrase; nothing emit-worthy yet. |
| 2 | in | READ | — | Functions as a preposition within a set phrase; unit still incomplete. |
| 3 | other | READ | — | Phrase remains incomplete; no stable unit. |
| 4 | words, | PARTIAL_SUMMARIZATION | 换句话说, | Idiomatic connective is complete with the current token; concise Chinese connective is appropriate. |
| 5 | we | READ | — | Subject present; predicate absent. |
| 6 | trained | READ | — | Verb present; object absent. |
| 7 | the | READ | — | Determiner begins an NP; incomplete. |
| 8 | language | READ | — | Inside the NP; still incomplete. |
| 9 | model | WRITE | 我们训练了语言模型 | Head noun closes the NP; the SVO clause is complete and safe to emit. |
| 10 | to | READ | — | Infinitive marker introducing purpose; verb pending. |
| 11 | classify | READ | — | Infinitive verb identified; requires object(s). |
| 12 | an | READ | — | Article; NP not yet formed. |
| 13 | abstract | READ | — | First object candidate; object remains open. |
| 14 | and | READ | — | Coordinator indicates multiple conjuncts; still open. |
| 15 | class | WRITE | 将摘要和类别分类 | Object closes with the current head; combine with "classify" to emit. |
| 16 | ah | DROP | — | Filler; drop. |
| 17 | to | READ | — | Preposition starting a new phrase; attachment unresolved. |
| 18 | abstract | READ | — | Repetition of earlier content; potential redundancy. |
| 19 | and | READ | — | Coordinator within the repeated span; still unresolved. |
| 20 | class | DROP | — | Local repetition ("abstract and class") established; drop as redundant. |
| 21 | ah, | DROP | — | Filler; drop. |
| 22 | if | READ | — | Introduces a condition; polarity and scope not yet determined, so no emission. |
| 23 | the | READ | — | Article; NP not yet formed. |
| 24 | abstract | READ | — | Head noun appears; complement still missing. |
| 25 | belongs | READ | — | Predicate present; complement pending. |
| 26 | to | READ | — | Preposition present; object missing. |
| 27 | the | READ | — | Article for the object NP; head not yet present. |
| 28 | class | READ | — | Object head present; condition's polarity still unspecified; hold. |
| 29 | or | READ | — | Coordinator signals an alternative; construction not closed. |
| 30 | not. | WRITE | ，无论摘要属不属于类 | Polarity is explicit; realize the condition with the compact "无论…" construction. |

Table 5: Step-by-step simultaneous translation actions for the sentence. Textual output: Here's my step-by-step sim of the simultaneous translation, with actions chosen to balance latency and quality (using DROP for fillers, PARTIAL_SUMMARIZATION to merge redundancy.

| Setting | BLEU | chrF | TER | COMET-da | COMET-KIWI | AL (s) |
|---------|------|------|-----|----------|------------|--------|
| Baseline (default prompt) | **63.87** | **45.94** | 139.66 | **0.8886** | **0.7903** | 2.120 |
| PARTIAL_SUMMARIZATION (↑) | 53.60 | 37.38 | 152.59 | 0.8745 | 0.7896 | **1.269** |
| SENTENCE_CUT (↑) | 49.29 | 34.52 | **137.93** | 0.8598 | 0.7707 | 1.322 |

Table 6: Effect of boosting one action's BLEU and lowering its AL in the inference prompt on the top-100 high-AL sentences.

to the provided per-action statistics. By preferring actions that appear more favorable in terms of the quality–latency trade-off, the model autonomously rebalances its strategy, demonstrating its capability to internalize external signals and to optimize interpretation decisions dynamically.

An example is provided below to show different decisions the LLM made when being informed by different statistics.

*Source sentence:* There has been a growing interest in the influence of English on other languages ah particularly ah related to English lexical borrowings, borrowings which sometimes have been called Anglicisms.

*Baseline:* 人们对英语对其他语言的影响的关注日益增加，尤其是与英语词汇借用有关——这种借用有时被称为"英语化"（Anglicisms）。

*PARTIAL_SUMMARIZATION↑:* 人们日益关注英语对其他语言的影响，尤其是与英语词汇借用相关的方面，这些借用有时被称为"英语化"。

In the translation process of this sentence, the second version utilized **PARI-TAIL_SUMMARIZATION** more often than the first version. As a result, the segment "There has been a growing interest in the influence of English on other languages" is translated more concisely with less word reordering. This helps improve the latency of this sentence remarkably.

A.8   LLM USAGE

This section describes the precise roles of large language models (LLMs) in our work, in accordance with the conference policy. LLMs were used both as components of our SiMT system and as general-purpose assist tools; they are not authors, and the human authors take full responsibility for all content.

**Models and versions.**   GPT-4o (`gpt-4o-2024-05-13`); Qwen3-8B; ChatGPT-5 (for editing).

**Roles in experiments.**

- **GPT-4o**:   Generated salami-based and action-adapted translations on the ACL60/60 English→Chinese and English→German dev sets.
- **Qwen3-8B**: Served as the base model for TransLLaMA supervised fine-tuning, few-shot learning, and DICL; also used for inference with both salami-based and action-adapted prompts.

**Roles in writing.**

- **ChatGPT-5**: Used strictly for copy-editing (grammar, wording, and minor style/formatting). It did *not* draft sections, introduce claims, or restructure arguments. All technical content, experiments, analyses, and conclusions were written and verified by the authors.

All model outputs (translations and edited text) were reviewed for accuracy; any errors were corrected by the authors. The authors accept full responsibility for the submission, including any content assisted by LLMs. LLMs are not eligible for authorship.

