# OpenReview forum: "Redefining Machine Simultaneous Interpretation: From Incremental Translation to Human-like Strategies"
_ICLR.cc/2026/Conference — ICLR 2026 Conference Withdrawn Submission_

### Official Review · Reviewer_MCYG · 2025-10-29

**Soundness:** 3
**Presentation:** 3
**Contribution:** 3
**Rating:** 6
**Confidence:** 3

**Summary:**

The paper introduces new adaptive actions to address strict time constraints in simultaneous machine translation: sentence_cut, drop, partial_summarization and pronominalzation. The actions enable restructuring of input and improve latency through summarization and dropping redundant information. These actions are similar to human interpreter strategies and are designed to modulate translation quality and latency. The approach is compared against salami-based segmentation, and are explored along inference-based approach (TransLLaMA) and prompting strategies such as dynamic in-context learning and few-shot prompting. A text-to-speech pipeline is developed to evaluate latency of the approach and baselines. Overall, interpretations based on salami approach have higher surface overlap, scoring higher in BLEU, chrF, and TER. Action-adapted references score better on semantic metrics (COMET). Further ablations reveal that the combination of sentece_cut and drop significantly reduce latency, and achieve the best tradeoff with improvement on semantic metrics.

**Strengths:**

1. The adaptive actions are carefully chosen to address specific challenges in simultaneous interpretation.
2. The evaluation pipeline is thorough and comprehensive.
3. The approach is tested with a range of inference and training methods.
4. Dynamic use of the actions in Table 3 shows significant improvement in quality and latency.
5. Good and clear analyses on the ablation studies reported in Table 2.

**Weaknesses:**

1. The writing occasionally lacks clarity. A few suggestions:
- It'd be great if there's an example demonstrating the difference between salami technique and sentence_cut
- The main results in Table 3 should emphasize flexibility/dynamicity of the action choices, and could be featured before Table 2.
- What model adaptation method is being used for the results in Table 2 and 3?
- Please clarify if multiple actions can be applied to the same input.
2. The method seems to work less well on En-De.
3. There seems like a missed opportunity without exploration on the distribution of action being chosen by the model.

**Questions:**

1. All of action choices could ideally reduce latency with shortened output. When applied individually in Table 2, however, why is latency increased in most cases? Can you address this in the paper?
2. Can there be multiple decision points in the same input? How may this be applied in practice?
3. Are there premature decisions that make interpretation worse? Can you provide an error analysis on these examples?

---

> ### Author Response · Authors · 2025-11-28
> **Response to Reviewer 3 (Part 1)**
>
> We thank the reviewer for taking the time to evaluate our paper and provide positive feedback. We address the reviewer’s questions as follows.
>
> Q1: All of action choices could ideally reduce latency with shortened output. When applied individually in Table 2, however, why is latency increased in most cases?
> A1: When compared with ACL60/60 reference latency (AL) scores, each of the individual actions can reduce latency. For example, in Table 2 (En-Zh), reference translations give the AL of 0.972s, while all individual actions result in less than 0.9 s AL.
>
> Q2: Can there be multiple decision points in the same input? How may this be applied in practice?
> A2: Yes, we can apply multiple actions in one input sentence. For example, “Sentence Cut + Drop”, “Drop + Partial Sum + Pronoun” and “All actions” in Table 2 mean that the LLM is allowed to choose from these actions when making step-by-step decisions. We also provide examples for the decision process of LLM to translate one input sentence in the appendix (A.6). The LLM can choose one action when an incremental word is input, just like the Read/Write decision making in conventional machine translation models.
>
> Q3: Are there premature decisions that make interpretation worse? Can you provide an error analysis on these examples?
> A3: Yes. We evaluated the COMET-KIWI score of every translation on ACL60/60 eval set, and the ACL60/60 references. Among all 416 sentences, 260 translations using our method is higher than the reference, while 148 are lower than the reference. Among these 148 sentences, we find some typical cases of inappropriate use of new actions that leads to the reduction of semantic similarity with the source sentences. (a) Overuse of summarization. For example, the reference translation for the source input “The execution of the node can be immediately started as a step.” is “节点的执行可以作为一个步骤立即开始”, and the translation produced by applying new actions is “节点执行可以立即开始”. The latter one omitted the input phrase “as a step”. Although this phrase is not important semantically, this overuse of summarization does not convey the exact meaning of the source sentence, and the words omitted are not truly redundant. This can be led by the lack of constraints when defining Partial Summarization action in the prompt, and LLMs’ nature of outputting shorter and more concise results.
> (b) Omission of conjunction words or pet phrases. For example, the reference translation of the input sentence “So, we tested two transformer based models” is “因此，我们测试了两个基于转换器的模型”, while our translation is “我们测试了两个基于变压器的模型”. The LLM applied “Drop” when dealing with the word input “So”, so this conjunction is omitted. Although this word is a common word filler during speech that conveys little actual meaning, it could represent the cause and effect meaning or act as a transition between two sentences, which is nontrivial and should not be dropped. This phenomenon can be resulted by the lack of restrictions of the definition of the action “Drop” in our prompt.

---

> ### Author Response · Authors · 2025-11-28
> **Response to Reviewer 3 (Part 2)**
>
> Regarding the weaknesses that you pointed out, we would like to do some clarification, which we hope could convey our intention and approaches more clearly.
>
> W1: It'd be great if there's an example demonstrating the difference between salami technique and sentence_cut.
> A1: We now show an example to illustrate the difference between salami technique and Sentence Cut action.
> Source sentence: A step that predicts a graph with complete structure and a select step that selects the nodes that are worth executing at this time.
> Salami technique (Definition: to divide the source sentence into minimum segments that contain enough information to translate)
> (A proposed step) (that predicts a graph with complete structure) (and a select step) (that selects the nodes) (that are worth executing at this time.)
> Full output: 一个建议的步骤，预测具有完整结构的图，和一个选择步骤，用来选择那些在当前时刻值得执行的节点。
> Sentence cut: apply the action Sentence Cut after reading the word “graph”. The sentence is cut into: A proposed step that predicts a graph. (It is) with complete structure and a select step that selects the nodes that are worth executing at this time.
> Full output: 一个建议步骤，用于预测一张图，它的结构很完整，还有一个选择步骤，用来选择当前值得执行的节点。
> The key difference between salami technique and Sentence Cut action is that, salami technique cuts the whole sentence into segments and use commas to connect them, while Sentence Cut only cuts when a long clause or a conjunction is recognized, and the LLM can use transition words (such as “it”, “and”) to connect the sentence segments.
>
> W2: What model adaptation method is being used for the results in Table 2 and 3?
> A2: Tables 2 and 3 do not involve any parameter-level model adaptation or fine-tuning. The backbone LLM is always frozen. For Table 2, we simply prompt the LLM to use a fixed action (or action combination) at each step and measure its BLEU and AL on the ACL60/60 dev set. We then treat these dev-set statistics as a form of prompt tuning: for Table 3 (on the ACL60/60 eval set), we augment the inference prompt with the BLEU/AL scores of each action and instruct the LLM to choose, at each step, the action that best balances quality and latency.

---

### Official Review · Reviewer_w1kE · 2025-10-31

**Soundness:** 1
**Presentation:** 1
**Contribution:** 2
**Rating:** 2
**Confidence:** 4

**Summary:**

This paper proposes expanding the set of adaptive actions available to simultaneous translation models by introducing SENTENCE_CUT, PARTIAL_SUMMARIZATION, DROP, and PRONOMINALIZATION. These actions allow the model to restructure sentences, omit less critical content, and simplify translations. By applying these adaptive actions, the models demonstrate increases in translation quality with a reduction in non-computationally aware latency.

**Strengths:**

•	The results section tests the approach across a wide variety of translation quality metrics and generally shows an improvement by applying the additional adaptive actions.

•	A solid ablation that identifies the influence of each of their proposed adaptive actions on downstream simultaneous translation quality and latency.

**Weaknesses:**

•	The approach feels more like a data preprocessing step that simplifies downstream translation rather than a fundamentally novel contribution. This raises concerns about fairness when comparing against baselines.

•	It is unclear whether modifications introduced by SENTENCE_CUT, PARTIAL_SUMMARIZATION, DROP, and PRONOMINALIZATION are always desirable. How can we verify that these changes maintain equivalence or acceptability compared to the original content?

•	The paper does not sufficiently justify the use of LLMs for simultaneous translation. Given their large parameter counts, wouldn’t they introduce latency that undermines real-time feasibility?

•	The full capitalization of adaptive actions is visually distracting.

•	Examples in Figure 2 are difficult to interpret without knowledge of the source language. Could the authors provide intermediate representations or choose an Indo-European language for better accessibility to English-speaking readers?

•	The work appears focused on text-to-text translation, yet includes components for speech-to-speech (e.g., TTS pipeline). The paper should either reframe as text-to-text or justify the need for speech-to-speech. If speech-to-speech is retained, a quality metric for speech output is necessary.

•	No computationally aware latency metric is provided to account for the additional time introduced by LLMs.

•	The paper does not compare against baseline encoder-decoder models.

•	Results are limited to a small number of datasets and language pairs.

•	Latency is measured only using Average Lagging (AL). Why not include LAAL, which accounts for overgeneration (a common issue with LLMs)? The justification provided (“it just serves as a reference of latency”) is insufficient.

•	Statistical rigor is lacking. Confidence intervals or averages over multiple runs would help demonstrate that improvements are not due to randomness.

**Questions:**

•	In Section 3.2, you mention four methods, but only three are described. Am I missing one?

•	Which language pair is used for the results in Table 1?

•	What do the ACL60/60 reference results in Table 2 represent?

•	Given the use of LLMs, is there a risk of information leakage that compromises the validity of simultaneous translation? Can you provide assurances that this is not affecting results?

---

> ### Author Response · Authors · 2025-11-28
> **Response to Reviewer 2 (Part 1)**
>
> We thank the reviewer for taking the time to evaluate our paper and provide positive feedback. We address the reviewer’s questions as follows.
>
> Q1: You pointed out that in Section 3.2, we mentioned four methods, but only three are described.
> A1: We described three major categories of methods: TranLLaMA, few-shot prompting and dynamic in-context learning (DICL). However, two approaches to search for the most suitable examples for each input sentence are applied in DICL, which are (1) keyword extraction and (2) semantic embedding clustering. They are mentioned in Section 3.2 lines 246-248, and described in detail in Appendix A.4. We counted these two approaches as two methods because they yield (significantly) different performances, as listed in Table 1.
>
> Q2: You asked about which language pair is used for the results in Table 1.
> A2: Table 1 listed the results on language pairs En-Zh. We apologize for this lack of clarification in the paper.
>
> Q3: You asked about what did the ACL60/60 reference results in Table 2 represent?
> A3: In Table 2, the “ACL60/60 reference” rows are simply the evaluation scores of the dev-set reference translations themselves. For form-based metrics (BLEU/chrF/TER), they are computed by comparing the references to themselves, so BLEU/chrF = 100 and TER = 0. For COMET-da and COMET-Kiwi, we evaluate the references against the source sentences, which gives the scores of the original high-quality offline translations and serves as a strong upper bound / baseline.
>
> Q4: You pointed out that information leakage might happen given the use of LLM in simultaneous machine translation.
> A4: We ran an additional experiment in which we fed the LLM incremental word sequences instead of full sentences. The final translations for each complete sentence were very similar to those obtained in our original setup. That said, we acknowledge this as a limitation: in future work, we plan to extend our system to accept incremental speech input rather than full text sequences, which will further reduce the risk of information leakage.

---

> ### Author Response · Authors · 2025-11-28
> **Response to Reviewer 2 (Part 2)**
>
> Regarding the weaknesses that you pointed out, we would like to do some clarification, which we hope could convey our intention and approaches more clearly.
>
> W1: You asked whether we can make sure that the introduced actions are always desirable. How can we verify that these changes maintain equivalence or acceptability compared to the original content?
> A1: We cannot guarantee that every individual use of new actions is desirable, but we do evaluate their overall effect with COMET-Kiwi, which measures semantic adequacy. Translations produced by our method achieve consistently higher COMET-Kiwi scores than salami-based translations, and higher scores than the ACL60/60 references in most settings (except En-Zh dev). On the ACL60/60 eval set, a sentence-level comparison shows that our action-based translations obtain higher COMET-Kiwi scores than the reference on 260/416 sentences, and lower scores on 148/416. The main reason for lower scores is lack of discourse context: the LLM only sees a single sentence, so when the correct interpretation depends on previous context, it cannot always match the context-aware reference translation.
>
> W2: You pointed out that given the large parameter counts in LLMs, they can introduce latency that undermines real-time feasibility.
> A2:We appreciate this concern. Our current work does not aim to build a full simultaneous MT system, but to propose a method for constructing training data that better reflects simultaneous behavior. In future work, we plan to combine our action-based policy with a more efficient backbone such as vLLM. The paper “LLMs Are Zero-Shot Context-Aware Simultaneous Translators” reports that a vLLM-based system can reach RTM=0.86, suggesting real-time feasibility; this indicates that, with an appropriate backbone, our method can be integrated into a truly real-time simultaneous translation system.
>
> W3: You asked for examples of the functions of new actions (Figure 2) in Indo-European languages.
> A3: We provide one example for each action in En-De language pair. The source and references are selected from ACL60/60 dev set, and the action-based translations are generated by GPT-4o.
> **Sentence_Cut**
> Source sentence: At the pool party with Barack Obama, we got a **graph** with the right nodes on the person and the event subject, but guessed the timing information wrong.
> Reference: Bei der Poolparty mit Barack Obama bekamen wir einen mit den richtigen Knoten zur Person und zum Ereignisthema versehenen **Graphen**, schätzten die zeitlichen Informationen aber falsch ein.
> Sentence_Cut: Bei der Poolparty mit Barack Obama bekamen wir **einen Graphen**. Er hatte die richtigen Knoten zur Person und zum Ereignisthema, aber wir schätzten die zeitlichen Informationen falsch ein.
>
> **Drop**
> Source sentence: These are the morphology level, **these are the morphology level** embeddings.
> Reference: Das hier ist die morphologische Ebene, **das hier sind die** Einbettungen auf der **morphologischen Ebene**.
> Drop: Das hier sind die Einbettungen auf der morphologischen Ebene.
>
> **Partial_Summarization**
> Source sentence: And here you have **the number of spans that were labeled as** English and **the spans that were labeled as other borrowings** and how many of them were unique.
> Reference: Und hier sehen Sie die Anzahl der Segmente, die als Englisch gekennzeichnet wurden, und der Segmente, die als andere Entlehnungen gekennzeichnet wurden, sowie wie viele von ihnen einzigartig waren.
> Partial_Summarization: Und hier sehen Sie, wie viele Segmente als Englisch oder als andere Entlehnung gekennzeichnet wurden und wie viele davon einzigartig sind.
>
> **Pronominalization**
> Source sentence: Lexical borrowing is a type of linguistic borrowing um which is basically reproducing in one language patterns of other languages. There are however some differences between **lexical borrowing** and code-switching.
> Reference: Lexikalische Entlehnung ist eine Form der sprachlichen Entlehnung, die im Grunde darin besteht, in einer Sprache Muster anderer Sprachen zu reproduzieren. Es gibt jedoch einige Unterschiede zwischen **lexikalischer Entlehnung** und Code-Switching.
> Pronominalization: Lexikalische Entlehnung ist eine Form der sprachlichen Entlehnung, die im Grunde darin besteht, in einer Sprache Muster anderer Sprachen zu reproduzieren. Es gibt jedoch einige Unterschiede zwischen **ihr** und dem Code-Switching.

---

> ### Author Response · Authors · 2025-11-28
> **Response to Reviewer 2 (Part 2)**
>
> Regarding the weaknesses that you pointed out, we would like to do some clarification, which we hope could convey our intention and approaches more clearly.
>
> W1: You asked whether we can make sure that the introduced actions are always desirable. How can we verify that these changes maintain equivalence or acceptability compared to the original content?
> A1: We cannot guarantee that every individual use of new actions is desirable, but we do evaluate their overall effect with COMET-Kiwi, which measures semantic adequacy. Translations produced by our method achieve consistently higher COMET-Kiwi scores than salami-based translations, and higher scores than the ACL60/60 references in most settings (except En-Zh dev). On the ACL60/60 eval set, a sentence-level comparison shows that our action-based translations obtain higher COMET-Kiwi scores than the reference on 260/416 sentences, and lower scores on 148/416. The main reason for lower scores is lack of discourse context: the LLM only sees a single sentence, so when the correct interpretation depends on previous context, it cannot always match the context-aware reference translation.
>
> W2: You pointed out that given the large parameter counts in LLMs, they can introduce latency that undermines real-time feasibility.
> A2:We appreciate this concern. Our current work does not aim to build a full simultaneous MT system, but to propose a method for constructing training data that better reflects simultaneous behavior. In future work, we plan to combine our action-based policy with a more efficient backbone such as vLLM. The paper “LLMs Are Zero-Shot Context-Aware Simultaneous Translators” reports that a vLLM-based system can reach RTM=0.86, suggesting real-time feasibility; this indicates that, with an appropriate backbone, our method can be integrated into a truly real-time simultaneous translation system.
>
> W3: You asked for examples of the functions of new actions (Figure 2) in Indo-European languages.
> A3: We provide one example for each action in En-De language pair. The source and references are selected from ACL60/60 dev set, and the action-based translations are generated by GPT-4o.
> **Sentence_Cut**
> Source sentence: At the pool party with Barack Obama, we got a **graph** with the right nodes on the person and the event subject, but guessed the timing information wrong.
> Reference: Bei der Poolparty mit Barack Obama bekamen wir einen mit den richtigen Knoten zur Person und zum Ereignisthema versehenen **Graphen**, schätzten die zeitlichen Informationen aber falsch ein.
> Sentence_Cut: Bei der Poolparty mit Barack Obama bekamen wir **einen Graphen**. Er hatte die richtigen Knoten zur Person und zum Ereignisthema, aber wir schätzten die zeitlichen Informationen falsch ein.
>
> **Drop**
> Source sentence: These are the morphology level, **these are the morphology level** embeddings.
> Reference: Das hier ist die morphologische Ebene, **das hier sind die** Einbettungen auf der **morphologischen Ebene**.
> Drop: Das hier sind die Einbettungen auf der morphologischen Ebene.
>
> **Partial_Summarization**
> Source sentence: And here you have **the number of spans that were labeled as** English and **the spans that were labeled as other borrowings** and how many of them were unique.
> Reference: Und hier sehen Sie die Anzahl der Segmente, die als Englisch gekennzeichnet wurden, und der Segmente, die als andere Entlehnungen gekennzeichnet wurden, sowie wie viele von ihnen einzigartig waren.
> Partial_Summarization: Und hier sehen Sie, wie viele Segmente als Englisch oder als andere Entlehnung gekennzeichnet wurden und wie viele davon einzigartig sind.
>
> **Pronominalization**
> Source sentence: Lexical borrowing is a type of linguistic borrowing um which is basically reproducing in one language patterns of other languages. There are however some differences between **lexical borrowing** and code-switching.
> Reference: Lexikalische Entlehnung ist eine Form der sprachlichen Entlehnung, die im Grunde darin besteht, in einer Sprache Muster anderer Sprachen zu reproduzieren. Es gibt jedoch einige Unterschiede zwischen **lexikalischer Entlehnung** und Code-Switching.
> Pronominalization: Lexikalische Entlehnung ist eine Form der sprachlichen Entlehnung, die im Grunde darin besteht, in einer Sprache Muster anderer Sprachen zu reproduzieren. Es gibt jedoch einige Unterschiede zwischen **ihr** und dem Code-Switching.

---

> ### Author Response · Authors · 2025-11-28
> **Response to Reviewer 2 (Part 3)**
>
> W4: You noted that our work is mainly text-to-text, but we include speech-to-speech components (e.g., TTS), and asked us to either reframe or justify this.
> A4: Our goal is to propose a method for generating training data better aligned with simultaneous text-to-text translation, not to build a full speech-to-speech system. We add a TTS step only to measure latency in seconds, which is the natural unit in real simultaneous translation. Following the spirit of *Simul-MuST-C* (which uses word-monotonicity–based metrics for speech), we convert text outputs to speech and compute second-level AL/LAAL to obtain a more realistic estimate of delay. Thus, the TTS pipeline is used purely for latency evaluation, not as evidence of a complete speech-to-speech system. A full evaluation of speech-to-speech quality (e.g., naturalness and intelligibility) is beyond the scope of this paper and is left for future work.
>
> W5: The paper does not compare against baseline encoder-decoder models.
> A5: Since we focus on text-to-text translation at this stage, we add a comparison with mBART as a baseline model, following the settings in *Simul-MuST-C*. We used our prompt of new actions to generate 25k translations (En-Zh) on MUST-C training datasets, and used MUST-C data as source sentences and our translations as references to train a mBART model. We evaluated the trained model on both ACL60/60 eval set and MUST-C tst-HE set. BLEU scores are 43.37 (ACL60/60 eval set) and 24.37 (MUST-C tst-HE). In comparison, using Qwen3-8B as the translation machine produced results with BLEU scores 62.44 (ACL60/60 eval set) and 34.43 (MUST-C tst-HE), which is significantly higher.
>
> W6: Results are limited to a small number of datasets and language pairs.
> A6: We add experiments on En-Ja language pair. Some important evaluation scores are as follows.
>
> | System    | BLEU  | COMET-Kiwi | LAAL (s) |
> |----------|:-----:|:----------:|:--------:|
> | Salami   | 27.23 |   0.8295   |  2.955   |
> | Action   | 33.73 |   0.8347   |  2.524   |
> | Cut+Drop | 33.06 |   0.8349   |  2.513   |
> | Reference| 100.0 |   0.8447   |  3.298   |
>
> On En-Ja, our action-based and cut+drop policies consistently outperform the salami baseline on both quality and latency. This supports the claim that our approach generalizes beyond En-Zh and En-De.
>
> W7: Latency is measured only using Average Lagging (AL). Why not include LAAL?
> A7: We re-evaluated all systems using LAAL, which explicitly incorporates the initial latency and is more robust and fair for comparison. Some main results are as follows.
>
> | Lang pair | Salami  | Action-based | Reference |
> |-----------|---------|--------------|-----------|
> | En–Zh     | 2.520 s | 2.281 s      | 2.907 s   |
> | En–De     | 1.116 s | 1.107 s      | 1.835 s   |
> | En–Ja     | 2.955 s | 2.924 s      | 3.298 s   |
>
> These results confirm that (i) the original AL numbers were not robust, and (ii) our conclusions remain valid under the more appropriate LAAL metric.

---

### Official Review · Reviewer_KaKM · 2025-11-01

**Soundness:** 2
**Presentation:** 1
**Contribution:** 2
**Rating:** 2
**Confidence:** 5

**Summary:**

This paper addresses the problem of trajectory synthesis for simultaneous machine translation (SiMT) . It extends the conventional *read/write* action space with four additional semantic actions:

(1) segmenting sentences into shorter semantic units,

(2) dropping non-informative words,

(3) summarizing partial prefixes, and

(4) replacing repetitive nouns with pronouns.

These enriched actions are used to construct training trajectories for adapting baseline models, including TransLlama and in-context learning (ICL) methods, which are then evaluated on the ACL 60/60 benchmark. The paper further leverages development-set statistics to directly prompt large language models to perform these enriched actions during SiMT inference.

**Strengths:**

The paper enriches the action space of SiMT to better emulate the behavior of human interpreters, introducing actions such as summarization, omission, and pronoun substitution. This is an intuitive and meaningful idea that bridges the gap between human interpretation strategies and machine translation policies, offering a more cognitively inspired approach to trajectory synthesis.

**Weaknesses:**

1. **Poor Writing and Presentation Quality:**

    - The font appears wrong.
    - In Figure 2, the actions are applied to the *reference translations*, yet Section 3.1 describes them as being applied to the *source text*, creating a contradiction.
    - The prompt design for ICL methods is not provided. Also, if *salami* references or other references are used as examples, this should be explicitly stated.
    - The meaning of “(a) reference translations” in line 254 is unclear; it seems to refer to using *TransLlama* style data synthesis with reference translations, but this should be clarified.
    - It is stated that model adaptation uses the ACL 60/60 dev set, but Tables 1 and 2 also report results on the dev set—does this imply training and evaluation on the same data, or is it a typo?
    - Line 264 “CozyVoice” should be “CosyVoice.”
    - In Section 3.4, it is implied that the proposed method is also used for direct SiMT inference, but this is never clearly described—this needs explicit explanation.
    - Tables 2 and 3 are oversized.
    - In Table 2 En-De, *TER* is a *lower-is-better* metric, yet the highest TER values are incorrectly highlighted.
    - The reported AL of the *reference translation* in Table 2 is conceptually unclear—how can the reference have latency below 1 s? Shouldn’t it be offline translation?
2. **Inappropriate Latency Metric:**

    The paper continues to use AL as the latency metric, despite known limitations. The community has largely moved toward Length-Adaptive Average Lagging (LAAL), which provides a more robust and fair comparison, as shown in recent IWSLT evaluations.

3. **Limited Effectiveness Across Language Pairs:**

    The proposed method shows improvement on **En–Zh**, but performs worse than *Salami* on **En–De** (Table 2). This suggests that the approach may not generalize well across languages. Additional experiments on more language directions are necessary to validate its overall effectiveness.

**Questions:**

Check weakness

---

> ### Author Response · Authors · 2025-11-28
> **Response to Reviewer 1 (Part 1)**
>
> We thank the reviewer for taking the time to evaluate our paper and provide positive feedback. We address the reviewer’s questions as follows.
>
> Q1: You pointed out that the prompt design for ICL methods is not provided.
> A1: Few-shot prompt and dynamic in-context learning (DICL) prompt
> [System]
> You are a professional English-to-Chinese simultaneous interpreter.
> Translate the given English sentence into Chinese.
> Do not output any thinking steps, only output the final result.
> Output only JSON: {"source": "...", "translation": "..."}.
>
> {FEW_SHOT_EXAMPLES}
>
> [User]
> Sentence:
> {SOURCE_SENTENCE}
>
> For FEW_SHOT_EXAMPLES,
> (1)Few-shot prompting experiment: we select 3 typical examples in each dataset (ACL60/60 reference, action-based translation, and salami-based translation on dev set), respectively. In Table 1, we listed evaluation scores of few-shot methods on these three datasets.
> (2)DICL experiment: we use 2 methods (keyword retrieval and classification; semantic embedding clustering) to find the best 3 examples in each dataset (ACL60/60 reference, action-based translation, and salami-based translation on dev set) for each input sentence.
>
> Q2: You suggested that the meaning of “(a) reference translations” in line 254 is unclear. Here are relevant sentences: All four methods are applied under three different development set configurations: (a) reference translations, (b) salami-based segmentation, and (c) action-based references.
> A2: In this section, both our few-shot prompting and DICL setups should be understood as prompt-tuning–style baselines: we keep the backbone LLM frozen and only change the prompts / in-context examples according to the three types of translation data (reference, salami-based, and action-based). In contrast, TransLLaMA is fine-tuned separately on each of these data types.
>
> The phrase “development set configurations” is indeed misleading. We will instead say “three types of translation data” and revise the sentence as: “All four methods are applied under three types of translation data: (a) reference translations, (b) salami-based translations, and (c) action-based translations.” Here, (a) are the off-line ¥human translations from the ACL60/60 dev and eval sets, (b) are translations obtained with the salami technique (using the prompt from Simul-MuST-C), and (c) are translations produced by our proposed action-based method (replacing “action-based references” with “action-based translations”).
>
> Q3: You pointed out that Tables 1 and 2 reported results on the dev set.
> A3: In line 323, we made a typo in the sentence “The evaluation results for ACL60/60 dev set are detailed in Table 1”. It should be ACL60/60 eval set. For Table 1, we trained / prompt-tuned the model using dev set, but did inference and evaluation on eval set. We sincerely apologize for the carelessness in making this mistake.
>
> For the method related to Table 2, our procedure is as follows. On the ACL60/60 dev set we systematically try different action combinations while keeping the LLM parameters frozen. This is a prompt-tuning setting: we only use the dev set to obtain, for each action, its BLEU and AL statistics, and then insert these statistics into the inference prompt. At test time on the eval set, the LLM sees these BLEU/AL numbers in the prompt and can, at each step, choose the action that best balances translation quality and latency.
>
> Figure 2 therefore reports the dev-set results for all action combinations. Besides providing the BLEU and AL statistics that we later reuse in the inference prompt, it also shows which action combinations achieve the best translation quality or the lowest latency.
>
> Q4: You suggested that in Section 3.4, it is implied that the proposed method is also used for direct SiMT inference, but this is never clearly described—this needs explicit explanation.
> A4: Our approach follows the way in paper “TransLLaMa: LLM-based simultaneous translation system”. LLM acts as a simultaneous interpretation machine in our experiments.

---

> ### Author Response · Authors · 2025-11-28
> **Response to Reviewer 1 (Part 2)**
>
> Q5: You indicated that the reported AL of the reference translation in Table 2 is conceptually unclear because they are below 1 second. You also suggested that we should use LAAL instead of AL for its robustness and fairness in comparison.
> A5: Thank you for pointing out this limitation. We agree that the originally reported AL values for the reference translations are conceptually unclear (and in some cases below 1s), because our AL computation did not account for the initial delay of wait-k systems. To address this, we re-evaluated all systems using LAAL, which explicitly incorporates the initial latency and is more robust and fair for comparison. Some main results are as follows.
>
> | Lang pair | Salami  | Action-based | Reference |
> |-----------|---------|--------------|-----------|
> | En–Zh     | 2.520 s | 2.281 s      | 2.907 s   |
> | En–De     | 1.116 s | 1.107 s      | 1.835 s   |
> | En–Ja     | 2.955 s | 2.924 s      | 3.298 s   |
>
> These results confirm that (i) the original AL numbers were not robust, and (ii) our conclusions remain valid under the more appropriate LAAL metric.
>
> Q6: You pointed out that although our method shows improvement on En–Zh, it performed worse than Salami on En–De (Table 2).
> A6: Thank you for this observation. To better understand the cross-lingual behavior of our method, we ran additional experiments on another language pair, En–Ja, on the ACL60/60 dev set, in parallel with Table 2:
> | System    | BLEU  | COMET-Kiwi | LAAL (s) |
> |----------|:-----:|:----------:|:--------:|
> | Salami   | 27.23 |   0.8295   |  2.955   |
> | Action   | 33.73 |   0.8347   |  2.524   |
> | Cut+Drop | 33.06 |   0.8349   |  2.513   |
> | Reference| 100.0 |   0.8447   |  3.298   |
>
> On En–Ja, our action-based and cut+drop policies consistently outperform the salami baseline on both quality and latency. This supports the claim that our approach generalizes beyond En–Zh. Our primary quality metric is COMET-Kiwi, which correlates better with human judgments and semantic adequacy than BLEU. Since our method intentionally moves away from the offline reference translations (which we argue are not ideal for simultaneous translation), a lower lexical overlap and thus lower BLEU is expected; what we care about most is preserving the source meaning under strict latency constraints, as reflected by COMET-Kiwi.

---

### Note · Authors · 2026-01-03

**Comment:**

We are withdrawing this paper to substantially revise the manuscript. We appreciate the reviewers’ and organizers’ efforts.

**Withdrawal Confirmation:**

I have read and agree with the venue's withdrawal policy on behalf of myself and my co-authors.